# Piperlongumine Induces Cellular Apoptosis and Autophagy via the ROS/Akt Signaling Pathway in Human Follicular Thyroid Cancer Cells

**DOI:** 10.3390/ijms24098048

**Published:** 2023-04-28

**Authors:** Tsung-Hsing Lin, Chin-Ho Kuo, Yi-Sheng Zhang, Pin-Tzu Chen, Shu-Hsin Chen, Yi-Zhen Li, Ying-Ray Lee

**Affiliations:** 1Department of Emergency Medicine, Kuang Tien General Hospital, Taichung City 433, Taiwan; drsixmg@gmail.com; 2Department of Hematology-Oncology, Ditmanson Medical Foundation Chia-Yi Christian Hospital, Chiayi 600, Taiwan; 05741@cych.org.tw (C.-H.K.); pintzu64@yahoo.com.tw (P.-T.C.); 3Department of Medical Research, Ditmanson Medical Foundation Chia-Yi Christian Hospital, Chiayi 600, Taiwan; newg7628@gmail.com (Y.-S.Z.); cathyhang1124@yahoo.com.tw (S.-H.C.); seraphim9515@gmail.com (Y.-Z.L.); 4Department of Microbiology and Immunology, College of Medicine, Kaohsiung Medical University, Kaohsiung 807, Taiwan; 5Master of Science Program in Tropical Medicine, College of Medicine, Kaohsiung Medical University, Kaohsiung 807, Taiwan; 6Faculty of Post-Baccalaureate Medicine, College of Medicine, Kaohsiung Medical University, Kaohsiung 807, Taiwan; 7Center for Tropical Medicine and Infectious Disease, Kaohsiung Medical University, Kaohsiung 807, Taiwan

**Keywords:** anticancer activity, apoptosis, autophagy, piperlongumine, thyroid cancer

## Abstract

Thyroid cancer (TC) is the most common endocrine malignancy. Recently, the global incidence of TC has increased rapidly. Differentiated thyroid cancer includes papillary thyroid carcinoma (PTC) and follicular thyroid carcinoma (FTC), which are the most common types of TC. Although PTCs and FTCs exert good prognoses and high survival rates, FTCs tend to be more aggressive than PTCs. There is an urgent need to improve patient outcomes by developing effective therapeutic agents for FTCs. Piperlongumine exerts anti-cancer effects in various human carcinomas, including human anaplastic TCs and PTCs. However, the anti-cancer effects of piperlongumine in FTCs and the underlying mechanisms are yet to be elucidated. Therefore, in the present study, we evaluated the effect of piperlongumine on cell proliferation, cell cycle, apoptosis, and autophagy in FTC cells with flowcytometry and Western blot. We observed that piperlongumine caused growth inhibition, cell cycle arrest, apoptosis induction, and autophagy elevation in FTC cells. Activities of reactive oxygen species and the downstream PI3K/Akt pathway were the underlying mechanisms involved in piperlongumine mediated anti-FTC effects. Advancements in our understanding of the effects of piperlongumine in FTC hold promise for the development of novel therapeutic strategies.

## 1. Introduction

Thyroid cancer (TC) is the most common endocrine malignancy. Because of the consequence of increased sensitivity of imaging modalities and pathologic identification of subclinical microscopic tumors, it has been one of the fastest growing cancers worldwide in the past few years. TC generally originates from follicular epithelial cells and is classified into well-differentiated, poorly differentiated, and anaplastic thyroid carcinomas [1,2,3]. Papillary thyroid carcinoma (PTC) and follicular thyroid carcinoma (FTC) are the subtypes of well-differentiated thyroid carcinoma, and FTC is less frequent but has a worse prognosis than PTC. [1]. The management strategy for PTC and FTC includes surgery for nearly all patients as well as radioiodine therapy and thyroid hormone therapy [1,2,3]. Although the prognosis of patients with differentiated thyroid cancer is generally good with treatment, those with metastatic and recurrent disease have limited treatment options and the 10-year survival rate is <10% [4]. FTC is the second most common type of TCs and counts for approximately 10–15% of all TCs. FTC tends to metastasize by hematogenous spread to distant sites, including bones and lungs, and is observed in up to 7–23% of FTC patients [5,6]. The long-term survival rates in patients with metastatic FTC are only 31–43% [5]. FTC is the major type of metastatic and recurrent TC in patients, which is identified as the most significant prognostic factor affecting survival in TCs. Therefore, it is crucial to develop an effective therapeutic agent for these patients.

Piperlongumine, an alkaloid extracted from *Piper longum* L., exhibits various bioactivities, including neuroprotective, antidepressive, antidiabetic, antiatherosclerotic, antimicrobial, anti-inflammatory, antiangiogenic, and anticancer activities [2,7,8]. Piperlongumine has been reported to suppress various human cancers, such as oral, head and neck, breast, lung, and liver and has been demonstrated to be nontoxic to normal cells [2,9,10,11,12,13,14,15,16,17,18,19,20,21,22,23,24,25]. Our previous report was the first to demonstrate that piperlongumine exerts anticancer activity in human papillary and anaplastic thyroid carcinoma (ATC) cells in vitro and in vivo [2]. Piperlongumine inhibits cell proliferation, induces cell cycle arrest, and elevates cellular apoptosis in TC cells. The underlying mechanisms of piperlongumine against PTC and ATC involve reactive oxygen species (ROS) induction and inhibition of the downstream Akt signaling pathway [2]. An in vivo study demonstrated that piperlongumine is capable of tumor inhibition and is safe for individual animals [2]. However, whether it can work against FTC needs further investigation.

In this study, the growth of FTC cells was inhibited with piperlongumine treatment. Cell cycle, tumor colony formation, cellular apoptosis, and autophagy were examined after this treatment. Additionally, the underlying mechanisms of autophagy were investigated. The results demonstrate that piperlongumine suppressed FTC proliferation and colony formation and induced cell cycle arrest, apoptosis, and autophagy via induction of ROS and suppression of the Akt signaling pathway. Furthermore, the administration of piperlongumine was shown to increase ROS and suppress the Akt signaling pathway in human FTC cells, thereby modulating cellular apoptosis and autophagy. Therefore, piperlongumine can be used as a therapeutic agent in patients with FTC.

## 2. Results

### 2.1. Piperlongumine Inhibits Cancer Cell Proliferation in Human FTC Cells

In our previous study, we demonstrated that piperlongumine exerts an anticancer effect on human papillary and anaplastic TC cells via cell cycle arrest and apoptosis; the underlying mechanism was found to be the ROS/Akt signaling pathway [2]. Here, we determined whether piperlongumine could inhibit cell proliferation in human FTC. WRO cells were used to verify cellular viability under piperlongumine treatment. The results showed that piperlongumine could suppress cell growth in in a dosage- and time-dependent manner (Figure 1A). Moreover, the half-maximal inhibitory concentrations (IC_50_) of piperlongumine in WRO cells were 10.24 μM and 5.68 μM at 24 h and 48 h post-treatment, respectively. To determine whether piperlongumine could reduce in vitro tumorigenesis, a colony formation assay was performed. WRO cells incubated with piperlongumine showed significantly less colony formation activity in a dosage-dependent manner (Figure 1B). As cell number reduction may be caused by cell cycle arrest and/or cell death, we checked the cell cycle of WRO cells under coincubation with piperlongumine. The results indicated that piperlongumine could induce cell cycle arrest in the G2/M phase in a dosage-dependent manner (Figure 1C), and the G2/M phage of WRO cells significantly increased from 22.2% to 32.1% and 18.1% to 32.0% at 6 and 12 h, respectively, after 10 μM piperlongumine treatment. Furthermore, flow cytometry detection of cellular apoptosis in WRO cells under piperlongumine treatment showed a significant increase in cellular apoptosis (Figure 2A). Western blotting was performed to investigate the underlying mechanisms of piperlongumine-mediated apoptosis in WRO cells. Decrease in pro-procaspase-9 and -3 and increase of cleavage caspase-9, caspase-3, and poly (ADP-ribose) polymerase (PARP) in a dosage-dependent manner were identified in the treated cells (Figure 2B). These data demonstrate that the administration of piperlongumine could suppress FTC cell growth as well as tumorigenesis and induce cell cycle arrest and apoptosis in vitro. Hence, piperlongumine is a potential novel therapeutic agent for FTC cells.

### 2.2. Piperlongumine Induces Cellular Autophagy, Autophagosome Formation and Autophagic Flux in Human FTC Cells

Our previous report showed that piperlongumine can increase cellular ROS and downregulate the downstream Akt signaling pathway, which is involved in cellular apoptosis induction [2]. In addition, activation of autophagy by piperlongumine via ROS has been reported [26,27]. Therefore, we evaluated the expression of LC3-II in WRO cells with piperlongumine treatment. The data confirmed LC3-II induction in WRO cells (Figure 3A), and this phenomenon was suppressed when combined with 3-methyladenine (3-MA) treatment (Figure 3A). Further investigation was conducted to examine autophagosome formation in the cells treated with piperlongumine. The results showed that piperlongumine increased autophagosome formation in WRO cells (Figure 3B). Moreover, rapamycin (an inducer of autophagy) was used as a positive control, and incubation with this drug also increased autophagosome formation and LC3-II induction (Figure 3A,B). Importantly, 3-MA acted as an autophagy inhibitor, and its administration to WRO cells under piperlongumine treatment blocked autophagosome formation (Figure 3B).

Because the elevation of LC3-II may cause the autophagy induction or blocking of cellular endogenous autophagic flux, in the present study, LC3-II was increased after piperlongumine treatment. To confirm whether piperlongumine could induce cellular autophagy, a pmRFP-EGFP-LC3 construction was transfected into WRO cells, and the autophagosome and autolysosome in the cells treated with piperlongumine along or in combination with chloroquine were examined with confocal microscopy. Figure 4 demonstrates that, compared to the control group, piperlongumine could induce autophagosome (yellow puncta) and autolysosome (red puncta) formation. Moreover, WRO cells treated with piperlongumine combination with chloroquine exerts autophagosome (yellow puncta) accumulation, suggested that incubation with piperlongumine could elevate autophagic flux in WRO cells. These findings suggest that piperlongumine induces autophagy machinery activation as well as autophagosome formation in FTC cells.

### 2.3. Piperlongumine Increases ROS, Which Contributes to Anti-Human FTC Activity

To determine the role of ROS in piperlongumine-mediated anti-FTC activity, the expression of cellular ROS was examined in WRO cells with or without piperlongumine treatment. The results demonstrated that piperlongumine induced ROS activity in the cells, and this phenomenon was blocked when cocultured with N-acetyl-L-cysteine (NAC) (Figure 5A). Further investigation was performed to determine the role of piperlongumine-induced ROS in anti-FTC behavior. Cell viability and colony formation were examined. The results show that blocking ROS with NAC in cells treated with piperlongumine reversed cell viability and colony formation compared with those incubated with piperlongumine only (Figure 5B,C). These results indicate that piperlongumine treatment increases cellular ROS and contributes to anticancer activity in human FTC cells.

### 2.4. Piperlongumine-Induced ROS Contributes to Autophagy Induction via the Akt Signaling Pathway in Human FTC Cells

The Akt, Erk, JNK, and p38 signaling pathways were studied to investigate the underlying mechanisms modulated by piperlongumine in WRO cells. Figure 6A shows that treatment with piperlongumine significantly reduced Akt activation, which was reversed when coincubated with NAC. However, NAC treatment only reversed the decreased *p*-Akt but not other MAPKs (Figure 6A). This finding suggests that piperlongumine induces ROS and further mediates Akt inactivation. Therefore, downstream factors, including mTOR, p70S6K, and the autophagic biomarker LC3, were examined with Western blotting. The results show that the activation of Akt/mTOR/p70S6K was suppressed in cells incubated with piperlongumine, which was correlated with LC3-II overexpression (Figure 6B). Previous reports have mentioned that ROS induction and inhibition of the PI3K/Akt/mTOR signaling pathway are two important pathways for autophagy induction [28]. Based on the above findings and our previous report [2], we speculated that piperlongumine mediates ROS induction and downstream Akt signaling inhibition to modulate cellular autophagy in FTC cells. To confirm this hypothesis, we evaluated the modulation of ROS in autophagy and autophagosome formation under piperlongumine treatment. The results reveal that the administration of piperlongumine activates the autophagic machinery. Blocking ROS induction with NAC in this situation significantly suppressed piperlongumine-mediated LC3-II overexpression (Figure 6C) and autophagosome formation (Figure 6B). Therefore, piperlongumine-induced autophagy appears to occur via an increase in ROS and the downstream Akt/mTOR/p70S6K signaling pathway in FTC cells. Hence, piperlongumine can be used as an effective anticancer agent for human FTC. The compound can suppress cell proliferation and tumorigenesis and induce cell cycle arrest, apoptosis, and autophagy via ROS induction and downstream Akt/mTOR/p70S6K inactivation. 

## 3. Discussion

PTC and FTC are the most common types of thyroid carcinomas. Although FTC is the second most common TC, FTC is more malignant than PTC. The prognosis of patients with advanced FTC who receive conventional treatment remains poor [4]. Therefore, anticancer strategies for patients with metastatic and recurrent disease have received increasing attention. In our recent study, we had demonstrated that piperlongumine can inhibit cell proliferation and induce cell cycle arrest and apoptosis by increasing ROS and downregulating the activation of Akt IHH-4, 8505c and KMH-2 cells [2]. In the present study, we evaluated the anticancer effects of piperlongumine in WRO cells. We assessed growth inhibition and tumorigenesis suppression as well as cell cycle arrest, apoptosis, and autophagy induction. The underlying mechanisms were further demonstrated. Our findings confirm that piperlongumine is a potential novel therapeutic agent in human FTC cells. Piperlongumine exerting cytotoxic activity with mutant p53 and targeting on Ras/PI3K/Akt/mTOR signaling pathway in human colon cancer cells has been reported [21,29]. The thyroid cancer cell lines IHH-4, 8505c, and WRO cells harbored BRAFV600E mutation [30,31], and the cells KMH-2 harbored NRASQ61R mutation [31]; therefore, whether the piperlongumine mediated anti-thyroid cancer cells activity was through these factors needs further investigation.

Incubation of human FTC cells with piperlongumine triggers growth inhibition and cell cycle arrest in the G2/M phase as well as intrinsic caspase-dependent apoptosis via ROS induction. These findings are similar to those of previous reports on human anaplastic TC and other types of human cancers [2,32,33]. ROS, a key mediator of cellular oxidative stress, plays an important role in cancer initiation and progression [33,34]. Recently, manipulation of ROS has emerged as a promising strategy for anticancer drug development [33,34,35]. Compared with normal cells, cancer cells usually harbor elevated ROS as a result of their uncontrolled proliferation and high metabolic rate, and the activity of ROS plays a vital role in cancer initiation and progression [33,34,35]. The high levels of ROS are toxic to tumor cells by causing oxidative damage to different cellular components, such as DNA, proteins, and lipids [36]. However, normal cells possess low basal ROS levels and high antioxidant activities [33,34,35]. Cells under the same pro-oxidant deviation and normal cells compared with malignant cells show tolerance, and minimal side effects are produced from this stress. Based on this hypothesis, piperlongumine has been reported to increase cellular ROS, which causes toxicity in human cancer cells but is safe for normal cells in vitro and in vivo [2,7,33,36]. In our previous study, we confirmed that piperlongumine can reduce tumorigenesis in vivo in anaplastic TC cells without significant toxicity in animals. Therefore, we speculate that piperlongumine can also be a safe therapeutic agent for anti-FTC in vivo; however, this hypothesis needs further investigation.

On the other hand, our study demonstrates that incubation with piperlongumine reduces the activation of Akt/mTOR/p70S6K signaling pathway in WRO cells (Figure 6B). Interestingly, it also reduces protein levels of Akt/mTOR and p70S6K partially, suggesting that treatment with piperlongumine may reduce these protein expressions of WRO cells. However, it needs further investigation to determine whether piperlongumine can modulate gene transcription and/or protein translation of Akt/mTOR/p70S6K.

ROS induction has been established to increase autophagy, and the underlying mechanisms, including NAD-dependent deacetylase sirtuin-1 (SIRT1), AMP-activated protein kinase, phosphoinositide 3-kinase/protein kinase B (PI3K/Akt), and mammalian target of rapamycin (mTOR), have been demonstrated [37,38]. In addition, suppression of autophagy induction or defect in autophagy leads to increased ROS in cells [39,40]. Autophagy is a self-clearance mechanism that delivers dysfunctional components to lysosomes for their degradation and maintains the physiological homeostasis of cells [41]. Therefore, autophagy plays a protective role in cells under stress. The process plays a dual role in cancer development, progression, and drug resistance [42]. In this study, we confirmed that incubation with piperlongumine can activate the autophagic machinery as well as autophagic flux and induce autophagosome formation in human FTC cells (Figure 3, Figure 4 and Figure 6C,D). Further investigations demonstrated that piperlongumine-mediated autophagy induction occurs via ROS elevation and the Akt/mTOR/p70S6K pathway (Figure 6C,D). Therefore, whether piperlongumine-mediated increase in autophagy plays a role in ROS degradation in FTC cells needs further examination. In addition, the role of piperlongumine-mediated increase in autophagy in anti-FTC cells must be explored.

Piperlongumine-triggered chemosensitivity in cancer therapeutics has been demonstrated in many studies [32,43,44,45]. In addition, autophagy exerting promotion of cancer cells via chemoresistance has been reported [46,47]. Our study demonstrates that the treatment of FTC cells with piperlongumine exerts an anticancer effect and causes increased cellular apoptosis and autophagy via the ROS/Akt pathway. Based on the above reports, the role of piperlongumine-mediated autophagy on FTC cells needs to be further explored.

Additionally, radioactive iodine therapy with radioiodine (131-I) is a common clinical treatment for PTC and FTC cases. Patients treated with radioiodine (131-I) produce ROS, which can induce oxidative stress with disturbance of redox balance and causes tumor cell cycle arrest and apoptosis [48,49]. Therefore, enhancing ROS production by pharmacological modulation is known to enhance radio response of cancers [36,49]. Importantly, piperlongumine increasing sensitivity of colorectal cancer cells to radiation through ROS induction has been reported [36]. Therefore, whether piperlongumine could promote radioiodine therapy in PTC and FTC is looking forward to further study.

On the other hand, piperlongumine has been demonstrated to synergistically enhance the antitumor activity of sorafenib by mediating ROS-AMPK activation and targeting CPSF7 in liver cancer cells [43]. Sorafenib and lenvatinib are multi-targeted tyrosine kinase inhibitors which are currently approved to treat advanced hepatocellular carcinoma, renal cell carcinoma, and radioiodine-refractory differentiated thyroid carcinoma [50]. Therefore, whether piperlongumine could enhance the therapeutic effects of sorafenib and/or lenvatinib in metastatic or radio-resistance human thyroid cancer cells also needs further investigation.

Autophagy is an evolutionarily conserved cellular process induced by nutrient starvation or lack of growth factors that clears unnecessary intracellular organelles and proteins to the lysosome for degradation [41]. Therefore, autophagy is also important in *protein* homeostasis, sequestering and degrading long-life proteins, and invading microbes [41,51,52,53]. Beside the classical degradative autophagy, a non-canonical function called secretory autophagy, which regulates unconventional secretory processes, is raising attention [54,55]. Secretory autophagy has been associated with the secretion of diverse proteins involved in cellular signaling, inflammation, and carcinogenesis [56,57]. Because secretory autophagy plays an essential role in cancer by sustaining cell proliferation, inhibiting apoptosis, enhancing angiogenesis and metastasis, immune cell regulation, modulation of cellular energy metabolism, and resistance to anticancer treatments are topics of significant clinical relevance [56,57,58]. Therefore, whether piperlongumine could modulate secretory autophagy in the WRO cells needs further investigation.

## 4. Materials and Methods

### 4.1. Cell Line and Cell Culture

The human FTC cell line WRO was kindly provided by Prof. Jen-Der Lin [59]. The cells were cultured in RPMI 1640 medium (GIBCO, Gaithersburg, MD, USA) supplemented with 10% fetal bovine serum (Biological Industries, Kibbutz Beit Haemek, Israel) in a 5% CO_2_ incubator at 37 °C. The passage range of the cells in this study was p05–p20.

### 4.2. Cell Viability Examination

The cells (5 × 10^3^ cells/well) were cultured in a 96-well culture dish with RPMI 1640 medium. The cells were allowed to attach overnight and were then incubated with the control medium (containing 0.01% DMSO) or piperlongumine (Cayman, Ann Arbor, MI, USA). Cell viability was verified with a CCK-8 assay kit (Enzo Life Sciences, Farmingdale, NY, USA) after 24 h or 48 h of treatment. Three independent experiments were performed.

### 4.3. Colony Formation Assay

To analyze the effects of piperlongumine on colony formation, the cells (10^3^ cells/well) were cultured in a six-well culture plate with RPMI 1640 medium. The cells were allowed to attach overnight and were then incubated with control medium or piperlongumine for 12 days. The cell colonies were stained with 10% crystal violet (Sigma-Aldrich, St. Louis, MO, USA) at 25 °C for 30 min. Colony formation analysis was performed as previously reported [60,61]. The size and the number of the colonies were determined three times.

### 4.4. Cell Cycle Analysis

The cells (1 × 10^6^ cells/dish) were seeded in a 10 cm cell culture dish and placed in a 37 °C incubator overnight. After attachment, the cells were incubated with control medium or piperlongumine under the indicated conditions. For cell cycle analysis, the cells were harvested and fixed with cold ethanol (70%) at 4 °C for 30 min after being washed with cold phosphate-buffered saline (PBS). Subsequently, the cells were washed with 50% ethanol, 30% ethanol, and cold PBS. The cells were stained with propidium iodide (PI; Sigma-Aldrich) containing RNase (Sigma-Aldrich) with PBS for 30 min at 4 °C. Finally, the cell cycle was examined with FACScan (Becton Dickinson, San Diego, CA, USA) and analyzed using the ModFit LT 3.3 software. Three independent experiments were conducted.

### 4.5. Cellular Apoptosis Determination

The cells (1 × 10^6^ cells/dish) were seeded in a 10 cm cell culture dish and placed in a 37 °C incubator overnight. The cells were cultured with control medium or piperlongumine, and cellular apoptosis was measured with Annexin V (Sigma-Aldrich) and PI staining. Flow cytometry was used to measure apoptosis as previously reported [62,63]. Proteins involved in piperlongumine-mediated cellular apoptosis were verified with Western blots. Primary antibodies, including caspase-9, caspase-3, and PARP, were purchased from Cell Signaling Inc. (Danvers, MA, USA). Glyceraldehyde 3-phosphate dehydrogenase (GAPDH) was used as a loading control, and the primary antibody was purchased from GeneTex (San Antonio, TX, USA). Three independent assays were performed.

### 4.6. Western Blotting

The cells (1 × 10^6^ cells/dish) were seeded in a 10 cm cell culture dish. After overnight attachment, the cells were incubated with control medium or piperlongumine for various durations. The cells were harvested and lysed using the M-PER^TM^ protein extraction reagent (Thermo Fisher Scientific Inc., Rockford, IL, USA) with a 0.1% protease inhibitor cocktail. The protein sample was harvested and segregated with sodium dodecyl sulfate-polyacrylamide gel electrophoresis (SDS-PAGE). The proteins were separated with SDS-PAGE and then transferred onto polyvinylidene fluoride (PVDF) membranes. The proteins were then examined with primary antibodies, including caspase-9 (Cell Signaling), caspase-3 (Cell Signaling), PARP (Cell Signaling), LC3 (Medical and Biological Laboratories, Nagoya, Japan), Akt (Cell Signaling), phosphor-Akt (Cell Signaling), Erk (Cell Signaling), phosphor-Erk (Cell Signaling), JNK (Cell Signaling), phosphor-JNK (Cell Signaling), p38 (Cell Signaling), phosphor-p38 (Cell Signaling), mTOR (Cell Signaling), phosphor-mTOR (Cell Signaling), P70S6K (Cell Signaling), phosphor-P70S6K (Cell Signaling), Beclin-1 (GeneTex), BNIP-3 (GeneTex), and GAPDH (GeneTex), at 4 °C overnight. The HRP-conjugated secondary antibody was added to the PVDF membrane at 25 °C for 1 h. Enhanced chemiluminescence reagent (Millipore, Burlington, MA, USA) was used to evaluate the protein bands. Three independent assays were performed, and one of the results is shown. Densitometry analysis was performed with ImageJ v1.53t.

### 4.7. Cellular ROS Determination

The cells (1 × 10^6^ cells/dish) were seeded in a 10 cm cell culture dish. After overnight attachment, the cells were treated with control medium or piperlongumine for 48 h. The cells were incubated with 2′,7′-dichlorodihydrofluorescein diacetate (DCFH-DA; 10 μM; Sigma-Aldrich) at 37 °C for 30 min. After washing with cold PBS, intracellular green fluorescence was analyzed with flow cytometry. The detail procedure was performed as previously reported [64,65]. NAC (Sigma-Aldrich) was used as an inhibitor of ROS. Three independent assays were performed.

### 4.8. Cellular Autophagosome Determination

The cells (1 × 10^6^ cells/dish) were seeded in a 10 cm cell culture dish with a cover slip and incubated in a 37 °C incubator overnight. After attachment, the cells were incubated with control medium, rapamycin (Sigma-Aldrich), 3-MA (Sigma-Aldrich), NAC, or piperlongumine for various durations. The autophagosomes were examined in the cells with immunofluorescent staining using LC3 primary antibodies (Medical and Biological Laboratories) and fluorescein isothiocyanate-conjugated secondary antibodies (GeneTex) at 4 °C overnight. The autophagosome formation was detected with laser confocal scanning microscopy (LSM800, ZEISS, Oberkochen, Germany). The nucleus was labeled with 4’,6-diamidino-2-phenylindole (Sigma-Aldrich). Autophagosome determination was performed as previously reported [66]. Three independent assays were conducted, and one of the results is shown.

### 4.9. Plasmid Transfection

To examine autophagosome and autolysosome formation in the cells treated with piperlongumine, the plasmid pmRFP-EGFP-LC3 (purchased from Addgene, Watertown, MA, USA) was transfected with Lipofectamine 2000 (Thermo Fisher Scientific) according to the manufacturer’s instructions. The autophagosome and autolysosome in the cells were determined as previously reported [66] and with laser confocal scanning microscopy (ZEISS) post-incubation with piperlongumine at 48 h.

### 4.10. Statistical Analysis

All data acquired from three independent experiments are shown as the mean ± standard deviation (SD). One-way analysis of variance and Fisher’s least significant difference test were performed with GraphPad Prism 7.0 (GraphPad Software, La Jolla, CA, USA). *p* < 0.05 represented a statistically significant difference.

## 5. Conclusions

In the present study, we have demonstrated that piperlongumine could significantly suppress human follicular thyroid cancer cell growth through cell cycle arrest at G2/M phase and cellular apoptosis as well as autophagy induction. Moreover, piperlongumine treatment elevated cellular ROS and further modulated Akt/mTOR pathway, which contribute to cellular apoptosis and autophagy. Our findings provide that piperlongumine is a potential drug candidate against incurable follicular thyroid cancer and is worthy of further clinical investigation.

## Figures and Tables

**Figure 1 ijms-24-08048-f001:**
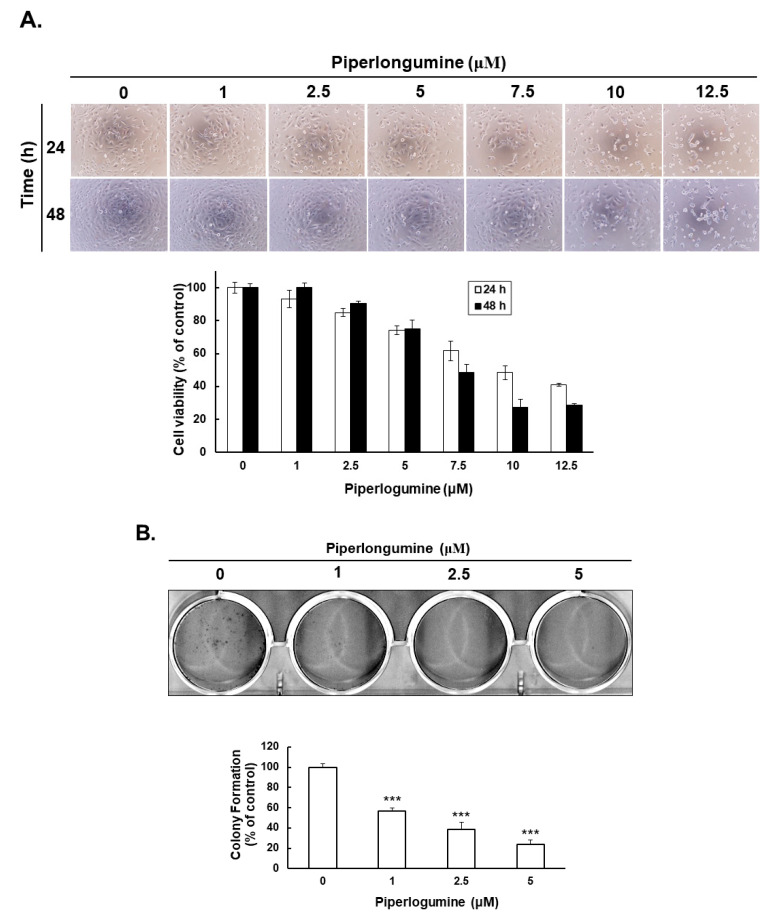
Piperlongumine suppressed cell proliferation and colony formation and induced cell cycle arrest in human follicular thyroid cancer cells. WRO cells were incubated with piperlongumine or control medium. (**A**) The cell viability was examined with microscopy and CCK-8 analysis. (**B**) The colony formation was determined after 12 days post-treatment. (**C**) The cell cycle regulation was examined as indicated, and the quantification data are shown on the right side. *** *p* < 0.001, compared with the control group.

**Figure 2 ijms-24-08048-f002:**
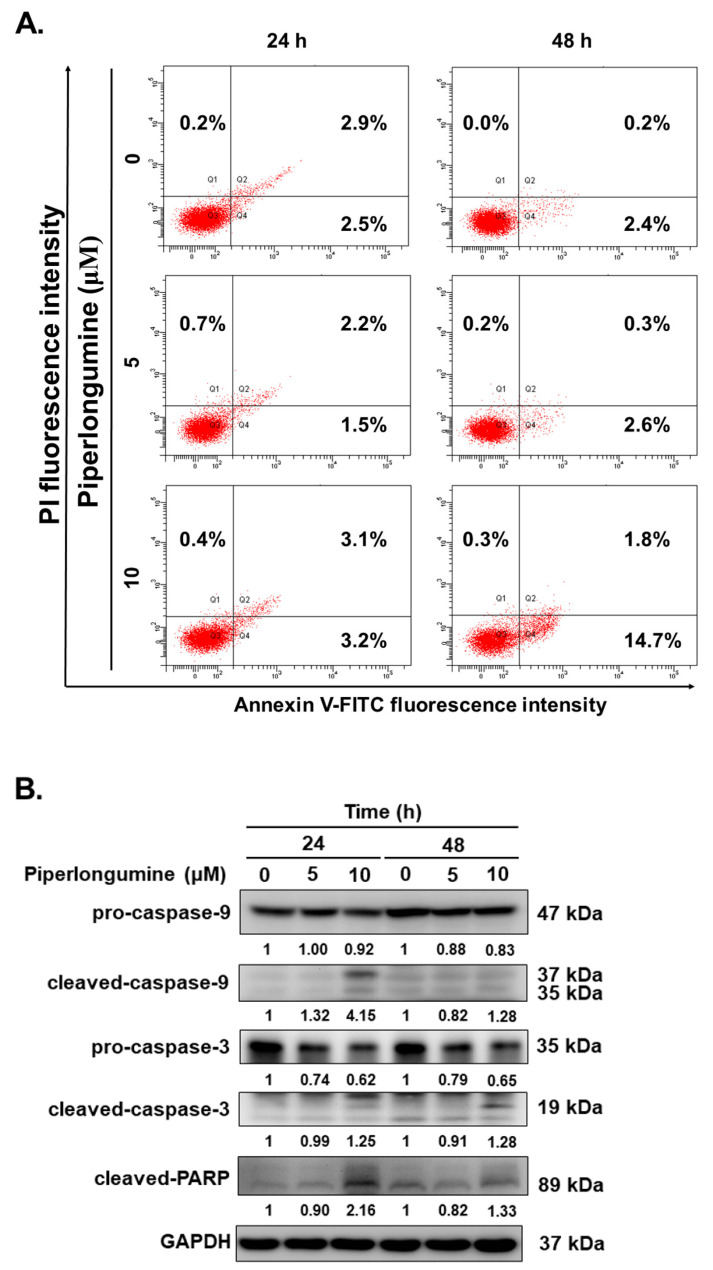
Piperlongumine induced intrinsic caspase-dependent apoptosis in human follicular thyroid cancer cells. WRO cells were incubated with piperlongumine or control medium. (**A**) Cellular apoptosis and (**B**) expressions of caspase-9, caspase-3, and PARP were determined with flow cytometry and Western blotting.

**Figure 3 ijms-24-08048-f003:**
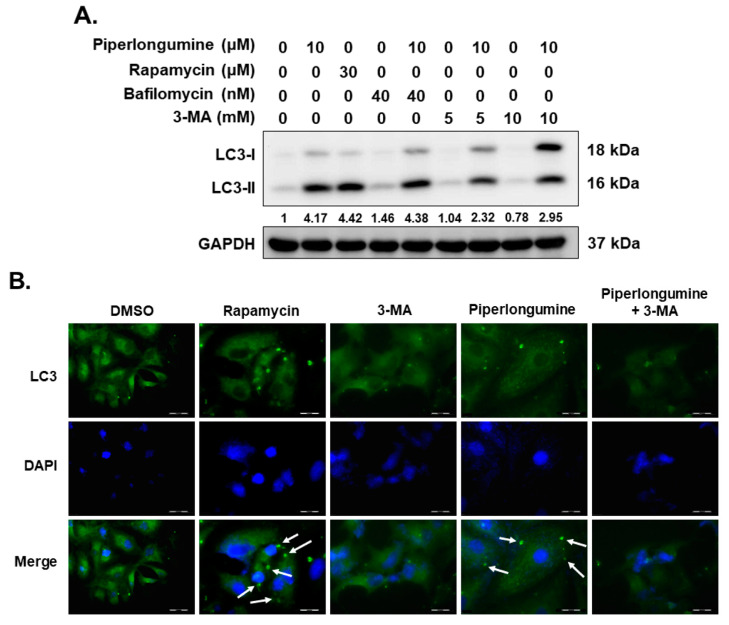
Piperlongumine induced cellular autophagy in human follicular thyroid cancer cells. WRO cells were treated with control medium, piperlongumine, and autophagic inducer or inhibitor, and the expressions of LC3-II were analyzed with (**A**) Western blotting or (**B**) immunofluorescent staining after 48 h. Rapamycin was used as an autophagic inducer, 3-MA was used as an autophagy inhibitor, and bafilomycin was used as an autophagic flux blocker. The arrow shows the autophagosome. The length of scale bars in (**B**) was 250 μm.

**Figure 4 ijms-24-08048-f004:**
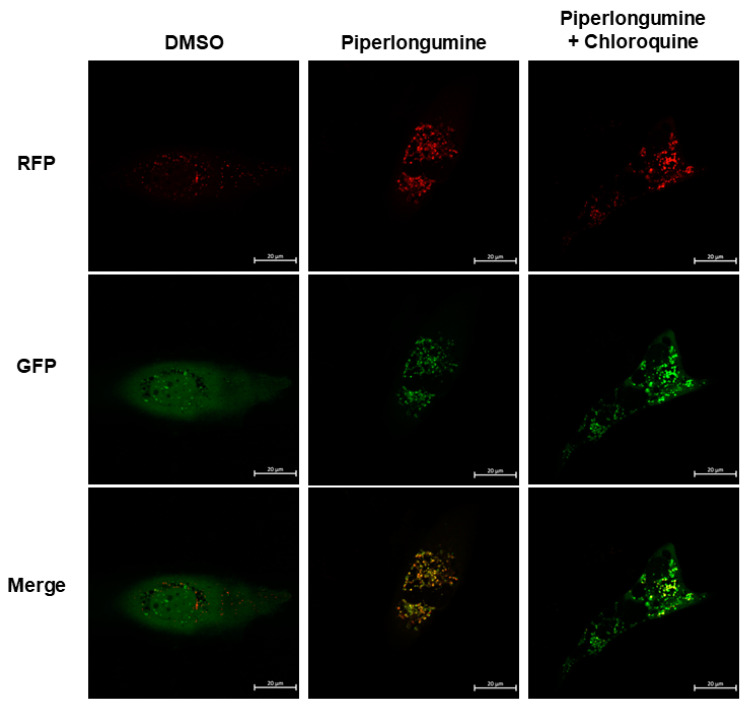
Autophagic flux in human follicular thyroid cancer cells following piperlongumine treatment. WRO cells expressing pmRFP-EGFP-LC3 were treated with piperlongumine (10 µM) for 48 h with or without chloroquine (5 µM) to examine autophagosome/lysosome fusion. Autophagosome (yellow) and autolysosome (red) puncta increased in the piperlongumine treated group, and most puncta were yellow (autophagosome) in the chloroquine combination group. Dimethyl sulfoxide (DMSO) was used as a negative control. The length of scale bars was 20 μm.

**Figure 5 ijms-24-08048-f005:**
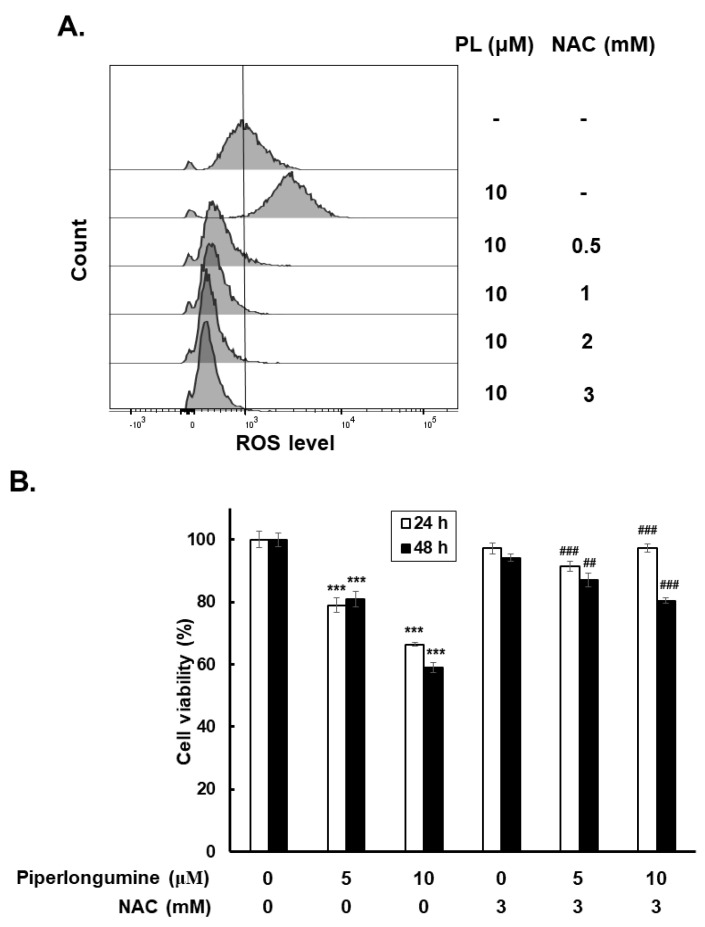
Piperlongumine increased cellular ROS and contributed to growth inhibition and colony formation suppression in human follicular thyroid cancer cells. WRO cells were incubated with control medium or piperlongumine as the descripted condition, and (**A**) intracellular ROS (**B**), cell viability, and (**C**) colony formation were determined. NAC was used as an ROS inhibitor. *** *p* < 0.001, compared with the control group. ^##^ *p* < 0.01, and ^###^ *p* < 0.001, compared with the piperlongumine-treated group. PL means piperlongumine. NAC means N-acetyl-L-cysteine.

**Figure 6 ijms-24-08048-f006:**
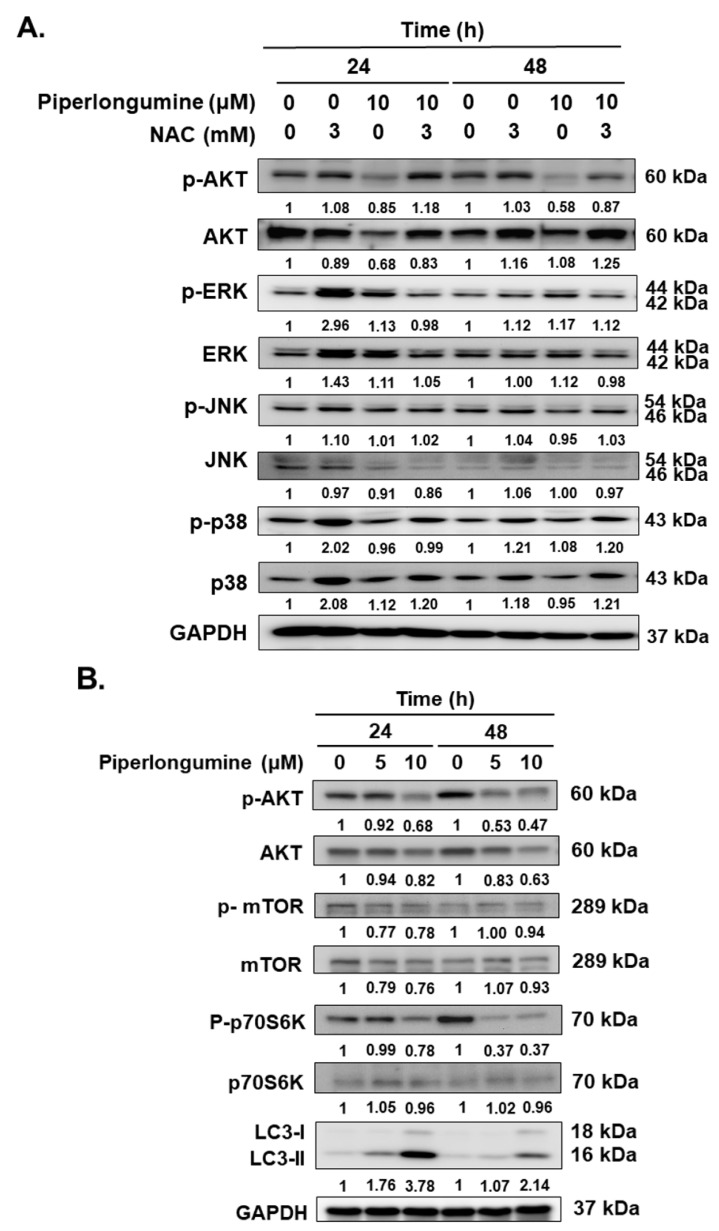
Piperlongumine modulated cellular autophagy via the ROS/Akt/mTOR/p70S6K pathway in human follicular thyroid cancer cells. WRO cells were incubated with control or piperlongumine, and (**A**) the activation of signaling pathways, including Akt, Erk, JNK, p38, and (**B**) mTOR, p70S6K, and LC3-II were examined with Western blotting. (**C**) Blocking of ROS with NAC in WRO cells with or without piperlongumine treatment, autophagy induction of LC3-II expression, and (**D**) autophagosome formation were detected with Western blotting or immunofluorescent staining after 48 h. The arrows present autophagosome puncta. The length of scale bars in (**D**) was 250 μm. NAC means N-acetyl-L-cysteine.

## Data Availability

Not applicable.

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
