# Peer review of "Piperlongumine Induces Cellular Apoptosis and Autophagy via the ROS/Akt Signaling Pathway in Human Follicular Thyroid Cancer Cells"

_ijms, 2023, doi:10.3390/ijms24098048_

Round 1

Reviewer 1 Report (New Reviewer)

Journal of International Journal of Molecular Sciences

Research Article;

The article entitled “Piperlongumine induces cellular apoptosis and autophagy via the ROS/Akt signaling pathway in human follicular thyroid cancer cells’’. The author investigated endocrine malignancy. As the incidence of thyroid cancer is increasing rapidly. Differentiated thyroid cancer includes papillary thyroid carcinoma and follicular thyroid carcinoma, is most common. Although both exert good prognoses and high survival rates. Piperlongumine exerts anti-cancer affects in various human carcinomas, including human anaplastic thyroid cancer. The author investigate the role in follicular thyroid carcinoma and the underlying mechanisms are yet to be elucidated. The effect of piperlongumine on cell proliferation, cell cycle, apoptosis, and autophagy in follicular thyroid carcinoma cells with flow cytometry and Western blot was observed by author which caused growth inhibition, cell cycle arrest, apoptosis induction, and autophagy elevation in follicular thyroid carcinoma cells. Activities of reactive oxygen species and the downstream PI3K/Akt pathway were the underlying mechanisms involved in piperlongumine mediated anti- follicular thyroid carcinoma effects.

I carefully read the manuscript and it needs revision for publication in the journal. I accept this article for publication after the revision. There are some common mistakes, figure, references and English language problem in the article which should be corrected by the authors. After the correction of all the mistakes, the article could be considered for publication in the prestigious International Journal of Molecular Sciences Journal.

Comments for Authors

Ø  The author needs to include latest reference in introduction section.

Ø  Manage the figure 1, figure 2 and figure 5 the author needs to revise and make it possible in one slide.

Ø  Write keywords in alphabetical order.

Ø  It will be batter to quantify western blot result in all manuscript. 

Ø  The needs to include graphical image of the study. 

Ø  Mentioned the original full image size in all Figures.

Ø  Write protein Kda in immunoblotting.

Ø  Revise Figure 5A,

Ø  In discussion section “the author needs to revised the discussion section, the author needs to focus on the study and don’t include irrelevant study.  

Ø  Check grammatically and spelling throughout the manuscript. There are many mistakes which needs to revise.

Cite the following references;

v  DOI: 10.2174/1871520622666220831124321

v  DOI: 10.1038/s41419-021-03771-z

Author Response

Reviewer 2 Report (New Reviewer)

The paper presents interesting findings and is worth publication in IJMS after minor revision.

In my opinion, the Introduction section should be expanded and the Conclusions is missed. Authors should add the list of abbreviations.

The sentence "Piperlongumine has been reported to suppress various human cancers, such as oral, head and neck, breast, lung, and liver, and has been demonstrated to be nontoxic to normal cells
[6,7,9,10,13,19-21,24,26,30,32,34,37,39,42,44,57]" is supported by too many citations. It should be explained in more details.

All other parts are well described by the authors. The methods, which were used by the authors are properly chosen to support their thesis.

Author Response

Reviewer 3 Report (New Reviewer)

The authors have evaluated the effect of piperlongumine in follicular thyroid carcinoma (WRO cell line), showing an activation of apoptosis and autophagy via ROS/Akt signaling pathway. The experiments are well designed, and the technical approaches are adequate. The main limitation of this study is to use only one cell line. In addition, some in vivo experiments should be performed, as previously analysed by authors with other cell lines (papillary thyroid cancer) (PMID: 34503074).

Author Response

Reviewer 4 Report (New Reviewer)

1. Introduction, Line 39 in, "Because the consequence of increased sensitivit" because must be changed to because of.

2. "Because the consequence of increased sensitivity of imaging modalities and pathologic identification of subclinical microscopic tumors, thyroid cancer has been one of the fastest growing cancers worldwide in the past few years"

This does not necessarily mean that there has been an actual increase in the number of people developing thyroid cancer, but rather that more cases are being detected due to advances in medical technology and diagnostic methods.

3. It is stated that TC is classified to "well-differentiated, poorly differentiated, and anaplastic thyroid carcinomas". Prognosis rate is really acceptable with current treatments for the first two. However, when it comes to anaplastic thyroid carcinomas, it is usually resistant to traditional forms of treatment and has a poor prognosis. Thus, studying the treatments for anaplastic thyroid carcinomas must be the field of interest.

4. "The underlying mechanisms of piperlongumine against PTC and ATC involve reactive oxygen species (ROS) induction and inhibition of the downstream Akt signaling pathway". ROS induction and inhibition of the downstream Akt signaling pathway are not equivalent. One is the subsequent of the other and it must be described in the text.

5. This work has no novelty at all. The previous work of the authors: https://www.ncbi.nlm.nih.gov/pmc/articles/PMC8428232/

has studied the anticancer effect of piperlongumine against IHH4, WRO, 8505c and KMH2 thyroid cancer cell lines in addition to in vivo IHH4 mouse model studies. This work is exactly the repeatition of part of the work against WRO cells which were already studied in the previous work. All studies are exactly similar:

-In this work Figure 1 A and B are dose-dependent cell viability( 24 and 48 hours) and colony formation in the range of 0-5 uM for WRO cells. In the previous work, dose-dependent cell viability( 24 and 48 hours) and colony formation in the range of 0-5 uM piperlongumine for WRO cells are shown in Figure 1 A and B, respectively.

-Figure 1C is cell cycle analysis after 6 and 12 hours at 0, 5 and 10 uM piperlongumine for WRO cells. In the previous study, Figure 2B shows cycle analysis after 6 and 12 hours at 0, 2 and 3 uM piperlongumine for WRO cells.

- Here, Figure 2A, represents Cellular apoptosis and 2B shows the expressions of caspase-9, caspase-3, and PARP after 24 and 48 hours at 0, 5 and 10 uM piperlongumine for WRO cells. In the previous study, the expression of caspases and PARP and apoptosis study are demonstrated in Figure 3. 

-ROS level with piperlongumine treatment in presence of different concentrations of NAC as ROS inhibitor is demonstrated in Figure 5. In the previous study, figure 4 shows ROS level with piperlongumine treatment in presence of different concentrations of NAC.

-Figure 6 represents the activation of signaling pathways, including Akt, Erk, JNK, p38, and mTOR. In the previous study, The expression and the activation of Erk, JNK, p38, Akt, and mTOR by piperlongumine is shown in Figure 6A.

The only data not already studied in the previous work, is autophagy induction.

Many previous studies have evaluated autophagy induction in cancer cells:

https://www.ncbi.nlm.nih.gov/pmc/articles/PMC6777669/

https://www.nature.com/articles/cddis2013358

https://www.nature.com/articles/bjc2013810

https://www.sciencedirect.com/science/article/pii/S1567576921002940

Overall, this work is just the repeatition of a small piece of the previous published study by these authors.

Author Response

This manuscript is a resubmission of an earlier submission. The following is a list of the peer review reports and author responses from that submission.

Round 1

Reviewer 1 Report

The authors investigated the molecular mechanism in the Piperlongumine-induced anti-proliferation effect in a follicular thyroid cancer cell line. The results indicated that piperlongumine could induce autophagy through ROS generation to suppress Akt/mTOR/p70S6K activation. Some revisions should be made.

Major points:

1. The expression of LC3-I in non-treated WRO cells of different results was not equal. For example, few in Fig. 3A and 5B, but easily observed in Fig. 5C.  The authors should make some explanations.

2. Fig. 2B: Cleaved caspase-3 should be presented, rather than pro-caspase-3 only, to support the induction of apoptosis.

3.  If autophagy is one of the reasons for the anti-proliferation effect of piperlongumine, treatment of 3-MA should also reverse the cytotoxic effect. Such data should be provided.

4. From Fig. 4A, 0.25 mM of NAC displayed significant suppression of ROS. Why the authors used 3 mM of NAC in the following experiments of cell viability and colony formation assay? Please make some explanations. 

5. Please add the quantification data of all western blots.

6. Please add the labels of each panel of Fig. 5D. (I guess they should be LC3 for the upper, DPAI for the middle, and merge for the lower panel).

7. For the descriptions of the results from Fig. 5A, the authors should emphasize that NAC treatment only reversed the decreased p-Akt but not other MAPKs.

Minor points:

1. The colonies were not easy to observe in Fig.1B. It is better to choose one well for each treatment and to enlarge the images.

2. The writing of AKT or ERK should be consistent (Akt or Erk was used in the legends of Figure 5).

Reviewer 2 Report

In this study, Ling et al. demonstrated that piperlongumine induced cellular apoptosis and oxidative stress while inhibiting Akt in human follicular thyroid cancer cells. However, whether it induced autophagy or inhibited autophagic degradation is not clear, as the autophagy flux experiment was not properly done. Moreover, the role of autophagy in the anticancer effect of piperlongumine remained completely unexplored.

1.      Introduction part of the abstract is too long, and no methods were mentioned.

2.      The colonies in the plate are poorly seen (Fig. 1B).

3.      What is the difference between the cell cycle on the left and right sides of Fig 1C? If it is the time of incubation, please mark it.

4.      It seems that piperlongumine inhibits autophagic degradation, as the LC3 II band in baflomycin+piperlongumine is not stronger than the LC3 II band in the bafilomycin sample. Actually, it seems that bafilomycin did not properly inhibit autophagic degradation, as it failed to increase LC3 II and p62 (Fig. 3A). Moreover, a substrate of autolysosomal degradation p62 was not weaker in piperlongumine than in the control sample. There is no bafilomycin in the text describing these results!!! Why did 3MA (inhibitor of autophagy induction) increase the concentration of p62 (inhibited degradation of autophagy) in the piperlongumine sample (Figures 3A, 6C)?

5.      Why was the same magnification not used for all the micrographs (Fig. 3B)? Why no autophagosomes were detected (LC3 fluorescent dots) in rapamycin-treated cells (not merged micrography)? On the other hand, LC3 fluorescent dots were clearly visible in 3MA-treated and control cells (although not indicated by arrows).

6.      It seems that NAC itself decreases the number of cancer colonies, but it could not be seen in the graph (Fig. 4C).

7.      Piperlongumine changes the concentration of total Akt, mTOR, S6K and MAP kinases (Fig. 5A and B). Please, discuss it.

Reviewer 3 Report

An investigation of the effects of the natural product piperlongumine against WRO thyroid cancer cells in vitro. This is a partial repeat of a previous study by the author using different thyroid cell lines (Kung et al., Cancers, 2021). This new manuscript is considerably weaker than the previous one and the data are not always properly interpreted.

1. Redundancy with the previous study. WRO cells have been used in the initial study [19]. Fig 1A is a repeat (copy!) of the previous study. IC50s were already reported. Nothing new here. This is not scientifically correct.

2.       Fig 1B. The top part, with the plate wells is superfluous. Cells cannot be seen. This image is totally useless. Remove it and keep the bar graph only.

3.       Line 97. It is concluded that the compound induces cell cycle arrest. Data in Fig 1C show almost no changes of the cell cycle profile. The conclusion must be reworked. The compound has very minor effects on cell cycle. Data are not properly interpreted, or in a biased form. This is unacceptable.

4.       Data in Fig 2 are very poor also, and again over-interpreted. The extent of PARP cleavage is very weak. A control drug, an established anticancer drug (doxorubicin, cisplatin) should be included as a comparator.

5.       The initial study included both in vitro (several cell lines) and in vivo (mouse model of thyroid cancer) data. Here the study includes only in vitro data (with a single cell line - no in vivo work). The novelty is very very limited.

6.       Altogether, this is a study with a single cell line (WRO), without control drug as a comparator, with redundancy compared to the initial study. The originality is weak and the data are often (not always) over-interpreted.

For these different reasons, I cannot support the publication of this manuscript. Other cell lines must be included and the induction of autophagy evidenced using an in vivo model.

Round 2

Reviewer 1 Report

Although the authors made some revisions to their manuscript, it still requires to be improved.

1. For cleaved caspase-3, please provide the images of long-exposed films.

2. The authors should include doxorubicin as a positive control for apoptosis markers (it is not necessary to use the same concentration as piperlongumine, just set as a positive control for the detection of apoptotic proteins!)

Author Response

Please seen the attachment.

Reviewer 2 Report

Q4. It seems that piperlongumine inhibits autophagic degradation, as the LC3 II band in baflomycin+piperlongumine is not stronger than the LC3 II band in the bafilomycin sample. Actually, it seems that bafilomycin did not properly inhibit autophagic degradation, as it failed to increase LC3 II and p62 (Fig. 3A). Moreover, a substrate of autolysosomal degradation p62 was not weaker in piperlongumine than in the control sample. There is no bafilomycin in the text describing these results!!! Why did 3MA (inhibitor of autophagy induction) increase the concentration of p62 (inhibited degradation of autophagy) in the piperlongumine sample (Figures 3A, 6C)?

Response Thank you for the comment. Currently, degraded autophagy and secreted autophagy are reported. Importantly, p62 exerts a cargo-receptor protein, makes the connection between the selected cargo and LC3 in the autophagosomal membrane (J Cell Biol. 2022, 221: e202110151). Therefore, it is not a particular marker of degraded autophagy. In addition, activation of autophagy by piperlongumine in biliary cancer cells and liver cancer cells has been demonstrated. In this study, WRO cells treated with bafilomycin blocking the degraded autophagy only elevated 1.46 times of LC3-II (comparing with control; Figure 3A), illustrating the level of endogenous autophagic degradation of WRO cells. However, treatment with piperlongumine increased 4.17 5 times of LC3-II comparing with control (Figure 3A). Therefore, we suggested that treatment with piperlongumine could induce autophagy in WRO cells

Q4. If there is autophagy induction LC3 II band in baflomycin+piperlongumine should be stronger than the LC3 II band in the piperlongumine sample. Appearance of the bands with calculated intensities (4.17 vs 4.38 without statistic) doesn’t seem much stronger. Moreover, a substrate of autolysosomal degradation p62 was not weaker in piperlongumine than in the control sample. If the authors think that decrease in p62 concentration in the piperlongumine sample cannot be seen as its synthesis was simultaneously upregulated, they should prove it. Anyway, bafilomycin A1 should increase p62 concentration.

Reviewer 3 Report

my previous comments have been taken into account. I can accept the answers. OK for publication.

Round 3

Reviewer 2 Report

If there is autophagy induction LC3 II band in baflomycin+piperlongumine should be stronger than the LC3 II band in the piperlongumine sample. Appearance of the bands with calculated intensities (4.17 vs 4.38 without statistic) doesn’t seem much stronger. Moreover, a substrate of autolysosomal degradation p62 was not weaker in piperlongumine than in the control sample. If the authors think that decrease in p62 concentration in the piperlongumine sample cannot be seen as its synthesis was simultaneously upregulated, they should prove it. Anyway, bafilomycin A1 should increase p62 concentration
